# PARALLEL RECURRENT DATA AUGMENTATION FOR GAN TRAINING WITH LIMITED AND DIVERSE DATA

## ABSTRACT

The need for large amounts of training image data with clearly defined features is a major obstacle to applying generative adversarial networks(GAN) on image generation where training data is limited but diverse, since insufficient latent feature representation in the already scarce data often leads to instability and mode collapse during GAN training. To overcome the hurdle of limited data when applying GAN to limited datasets, we propose in this paper the strategy of *parallel recurrent data augmentation*, where the GAN model progressively enriches its training set with sample images constructed from GANs trained in parallel at consecutive training epochs. Experiments on a variety of small yet diverse datasets demonstrate that our method, with little model-specific considerations, produces images of better quality as compared to the images generated without such strategy. The source code and generated images of this paper will be made public after review.

## 1 INTRODUCTION

*Generative Adversarial Networks*(GAN)(Goodfellow et al. (2014)) are powerful unsupervised learning models that have recently achieved great success in learning high-dimensional distributions in various types of problems and on different datasets. In the context of image generation, the basic framework of a GAN model consists of two parts: a generator $G$ that generates images by translating random input $z$ into an image, and a discriminator $D$ which determines the authenticity of a generated image $\mathbf{x}$ as compared to the real data. These two components are alternatively optimized against each other during the training process, with the goal of minimizing the difference between the distribution of generated image data and target distribution of real image data.

A notable challenge in GAN training, however, lies in the need for large amounts of clearly labeled data to capture the diversity features across various types of images into the model. Such requirement makes it difficult or even impossible to utilize GAN in applications where the amount of available training data is small but diverse. Moreover, recent deep learning models (Goodfellow et al. (2015)) have demonstrated tendencies of misrepresentation in classification tasks when influenced by adversarial noise. Such vulnerability may also translate to unsatisfactory image generation as most generative models are implemented with deep networks.

Thus, given these considerations, we propose in this paper the strategy of *parallel recurrent sample augmentation* agnostic to specific model details. Our contributions can be summarized as follows:

- We proposed a general black-box method using recurrent image addition to diversify training data and enhance its quality over a large class of GANs without model specifications.

- We also includes in our model a novel $K$-fold parallel framework, which better augments training data by stabilizing model output and preventing overfitting.

- Experiments across various datasets and GAN objectives demonstrate the effectiveness of our method using authenticity measures such as Inception Score and Frechet Inception Distance.

## 2 RELATED WORK

Building reliable deep generative adversarial models on limited amounts of training data has been a persistent challenge within the research community. Previous efforts to address the issue of labeled data scarcity generally fall into two groups: optimizing the structures of GANs to allow for better feature representation of data, and augmenting the training data through techniques.

Along the first line of research, prior research optimized the GAN in Goodfellow et al. (2014) by considering stronger mathematical objectives for more powerful latent space representation in general(Nowozin et al. (2016), (Arjovsky et al. (2017),Gulrajani et al. (2017), Mroueh & Sercu (2017)). In addition, recent research on GANs (Gurumurthy et al. (2017),Nowozin et al. (2016)) reparametrized the input noise using variational inference(Kingma & Welling (2014)) by assuming that the latent space could be modeled by a tractable prior, but noise reparametrization has severe mathematical limitation that prevents applicability to more general models. Furthermore, distributed multi-discriminator models(Intrator et al. (2018),Durugkar et al. (2016)) also enhance the performances, with great potential room for further optimization.

For the second line of research, data augmentation has already enjoyed considerable success in the problem of image classification(Krizhevsky et al. (2012)). Traditional data augmentation methods such as crop, mirror, rotation and distortion (Krizhevsky et al. (2012),Simard et al. (2003)) generally require domain-specific expert knowledge and manual operations, and only produce limited variation in augmented images. Recent developments centered on automatic augmentation of training data using controlled RNNs after transforming the problem into policy search in reinforcement learning (Cubuk et al. (2018)) , but the space complexity the search algorithm requires still does not apply to quick augmentation with limited data.

## 3 PROPOSED METHOD

In this section we describe the details of Parallel Recurrent Data Augmentation(PRDA), which consists of recurrent image data construction via noise addition on images and parallel generation with fold division.

### 3.1 RECURRENT DATA AUGMENTATION

Recent research suggests that adversarial noise to deep learning models greatly perturbs the performances of deep neural networks. In CNNs for classification, for instance, the neural network may assign irrelevant labels to semantically unambiguous images after the addition of background noise(Goodfellow et al. (2015)). This phenomeon is due to the omission of possible latent features in new data generation caused by over-dependency on our limited training data. Since virtually all GANs are implemented with deep networks, a similar deficiency in representation of latent feature space may translate to lower qualities of the generated images.

To counter this effect, we consider the strategy of *recurrent data augmentation*, which constructs varied images given the limited training set by repeatedly generating and modifying these samples on training set for subsequent training sample generation. Running the original generative model for a fixed number of times, we extract sampled images using standard procedures of sample image generation as described in Radford et al. (2015),Gulrajani et al. (2017). Random noise is then added to these samples to produce new images, which are then used for subsequent training. This procedure is repeated for a fixed number of times or until convergence. Figure 1 is a flow-chart of our procedure.

Notice that the addition of high dimensional normal random noise allows the additional images to retain the original latent features to be learned by GAN. Compared with traditional methods such as rotation, cropping and mirroring (Simard et al. (2003),Krizhevsky et al. (2012)) which may lead to information loss, random noise addition doesn't reduce the information about the latent features in training set while making the model more robust, because the expectation of noise is invariant at 0. Additionally, noise addition is agnostic to the type of generative model, since the procedure is independent from the specific choice of neural network or objective functions.

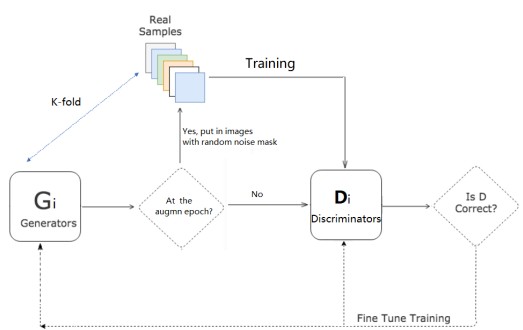

Figure 1: Recurrent Data Augmentation

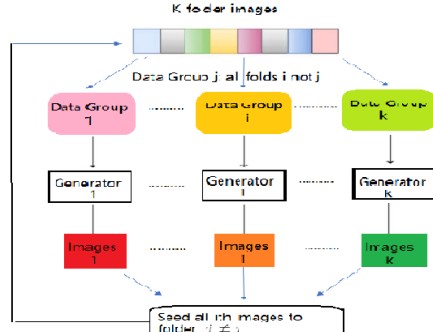

Figure 2: Parallel Image Generation

## 3.2 PARALLEL IMAGE GENERATION WITH FOLD DIVISION

Additionally, we introduce a parallel data generation strategy inspired by $K$-fold cross validation in machine learning(Bishop (2006)). Dividing the training data into $K$ folds at the beginning, we run in parallel $K$ independent generators on $K$ data groups, each consisting of $K - 1$ folds of the training set. When data is generated in each generator, the sample images produced by each generator at the given epochs are then added with random noise. These noised images, in turn, are fed back into the respective training data sets. To allow for maximal usage of each generated image, we insert the images such that the image generated by one generator goes to the augmented training set of all other $K - 1$ generators. This is to insure that the different generators in parallel have access to as many varied data pieces as possible in subsequent steps of training, so as to prevent overfitting and bolster the robustness of our model. Figure 2 demonstrates the mechanism of our algorithm.

Notice that our $K$-fold division goes hand in hand with the recurrent data generation with no need for model specific considerations. As demonstrated by our experiments Section 4, training different GANs in parallel from different folds of data substantially boosts the quality of the training set and that of the generated images.

## 4 EXPERIMENTS

For a comparative analysis, we have conducted experiments on previous GAN architectures over various datasets with/without data augmentation. The GANs we have tested on include DC-GAN( Radford et al. (2015)), BEGAN(Berthelot et al. (2017)), and WGAN-GP(Gulrajani et al. (2017)). Additionally, to simulate limited data, we randomly select 5000 images from the datasets CIFAR-10, CelebA and Places to create their corresponding reduced datasets named reduced-CIFAR, reduced-CelebA, and reduced-Places, and conduct our experiments on these limited datasets. All of our experiments are conducted with CPU Intel(R) Core(R) CPU 8700-K (3.7GHz) and GPU GTX 1080.

### 4.1 QUALITATIVE EVALUATION

In our experiments, we augment the training set with 8 noised images every 100 training epoches, and repeat the procedure 3-5 times. By comparison, the unaugmented GAN is run over the same initial training data, with the number of epochs the same as the product of 100 and augmentation times. Figure 3,4,5 are some sample images that our method produces with the state-of-the-art GAN WGAN-GP as compared to the ones produced by GAN without data augmentation. We observe that GANs with parallel recurrent image augmentation produce semantically coherent and visually diverse images earlier than the unaugmented GANs, while able to avoid fluctuations seen in unaugmented GANs during training.

### 4.2 QUANTITATIVE EVALUATION

To evaluate the quality of the images generated by our augmentation method as compared with those generated without augmentation, we use the Inception Score(IS)(Salimans et al. (2016)) and Frechet

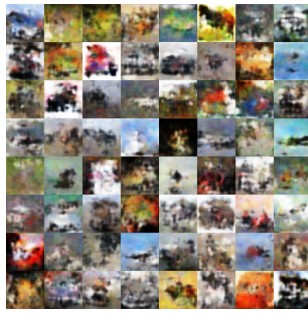 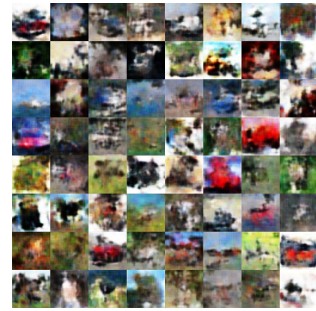 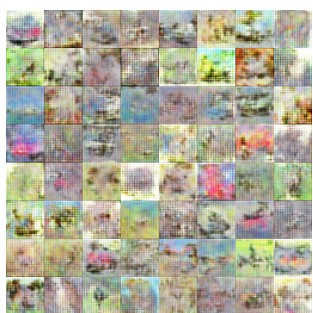

Figure 3: Augmented GAN at Epoch 271,Reduced-CIFAR    Figure 4: Unaugmented GAN at Epoch 361,Reduced-CIFAR    Figure 5: Unaugmented GAN at Epoch 342, Reduced-CIFAR

Inception Distance(FID)(Heusel et al. (2017)). IS measures the entropy of generated images, with higher scores indicating greater diversity. On the other hand, FID measures the distance between the generated data and real data with two respective means and variances. Thus, the larger the IS and the smaller the FID, the better the performances of the model.

The inception score(IS) and Frechet Inception Distance (FID) we used in all experiments are calculated by Python scripts available respectively at `https://github.com/openai/improved-gan/tree/master/inception_score` and `https://github.com/bioinf-jku/TTUR`.

| Datasets | K=3 | K=5 | K=10 | No Augmentation |
|---|---|---|---|---|
| Reduced-CelebA | 2.75/179.9 | **2.96**/161.8 | 2.49/**123.2** | 2.33/203.5 |
| Reduced-CIFAR | 2.33/133.6 | **2.45/116.3** | 2.28/132.2 | 1.91/265.7 |
| Reduced-Places | 2.29/**175.8** | **2.41**/181.8 | 2.12/193.1 | 1.87/242.1 |

Table 1: IS/FID scores with different $K$ values on different datasets, using WGAN-GP

| GANs | Augmented | Not Augmented |
|---|---|---|
| DCGAN | **1.76/213.3** | 1.69/267.8 |
| BEGAN | **1.93/205.7** | 1.74/261.3 |
| WGAN-GP | **2.45/116.3** | 1.91/265.7 |

Table 2: IS/FID scores of GANs on Reduced-CIFAR with/without Augmentation, given K = 5

Table 1 lists the combinations of GAN and dataset we tested our strategy on, as well as the Inception Score and Frechet Inception Distance of the images that are generated with and without our method using the state-of-the-art GAN WGAN-GP. Table 2 lists IS and FID of different GAN models on the Reduced-CIFAR with/without data augmentation. Clearly, on a variety of GAN structures and Datasets, recurrent sample augmentation produces better images as measured quantitatively.

## 5   CONCLUSION AND FUTURE WORK

In sum, our paper shows that parallel recurrent sample augmentation can significantly improve the quality of synthetic images for a large class of GAN models. Our strategy is not only simple to implement, but also agnostic to the specific type of GAN to be improved on.

As a further step, we are investigating the relationship between our proposed approach and other established methods. One possible pathway, for instance, lies in reinforcement learning as described in Cubuk et al. (2018) that gives more control to image generation via reward designation. We also hope to apply our idea to other generative models such as the VAE(Kingma & Welling (2014)) and further optimize our strategy using recent theoretical advances.

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
