# OpenReview forum: "Parallel Recurrent Data Augmentation for GAN training with Limited and Diverse Data"
_ICLR.cc/2019/Workshop/LLD — Submitted to LLD 2019_

### Official Review · AnonReviewer1 · 2019-04-05
**Too poor technical contribution to the LLD field**

**Rating:** 1
**Confidence:** 3

**Review:**

The authors presented what they called a "parallel recurrent data augmentation" technique for increasing the training images available for learning GANs. The method consists in introducing random noise to input images in a similar to K-fold cross-validation fashion: K GANs are trained simultaneously using different portions of the original training data, and the images generated by the K-th generator are perturbed with random noise and included in the training subsets of the other K-1 GANs. The method is evaluated on a simulated limited data experiment using CIFAR-10 images, showing improvements in terms on standard evaluation metrics in generative modelling.

In my opinion, the technical contribution is too weak for accepting the paper. Introducing random noise for data augmentation is standard not only in the GAN literature but also in deep learning in general.

The idea of feeding multiple generators with the artificial outputs of a surrogate models is perhaps novel but its impact is limited and it is not properly explored in this submission. The quantitative contribution in the results cannot be attributed to this technique, also, as it could be maybe just the consequence of adding noise to the inputs. In this sense, the paper lacks a comparison with respect to feeding the generators with noisy images from the training set.

It would be also important to show if the parallel strategy has some contribution to avoid standard GANs issues such as mode collapse.

The data set used for evaluation (CIFAR-10) has a significantly low resolution, and therefore it is difficult to see if the semantics of the images are actually preserved. The article would benefit from including either a larger resolution data set or at least an experiment showing e.g. that the classification performance of a DNN trained for CIFAR-10 image classification using only artificial samples doesn't differ too much from a similar network trained on a real CIFAR-10 sample.

The article also suffers from some presentation issues. The related works section describes GANs research and data augmentation, but without focusing on the problem that is intended to be solved with the propose tool (learning GANs with limited data). Section 2 should be reorganized to include important citations regarding this issue and the existing alternatives to solve it.

Other minor comments:
- There are spaces missing between words and parenthesis, specially with most of the citations.
- The first sentence in the abstract is too long and redundant.
- Second paragraph in Section 2 suffers from many repetitions of the word "research".
- Figure 2 is unreadable due to poor resolution.

---

### Official Review · AnonReviewer2 · 2019-04-05
**An interesting idea, but needs some polish**

**Rating:** 2
**Confidence:** 1

**Review:**

Summary:
The authors propose a method for improving the image generation quality of GANs by (1) augmenting the training set with noise-enriched samples and (2) running multiple GANs in parallel over different subsets of the data and periodically augmenting their respective trainings with images sampled from the other GANs.

I find the second contribution more compelling than the first. It is a clever augmentation strategy that does not require human intervention. Overall, however, the presentation hinders the conveyance of the idea enough that I don't think this work is ready yet for release. With a little more polish clarifying the ideas and cleaning up the writing/figures, I think it could be an excellent submission to another workshop or conference down the road.

- It is unclear how much of the reported gains are due to the first contribution vs the second.
- The first sentence of the abstract introduces too much too quickly.
- I'm not sure what point you're trying to make with the comment about "demonstrated tendencies of misrepresentation"
- There are many issues with the writing: spacing issues (especially around parentheses), and minor grammar oddities
- Missing related work in your mention of automatic augmentation: TANDA (Ratner 2017);
- It is unclear to me why rotation and mirroring may lead to information loss, as you claim
- The text in Figure 2 is very hard to read
- The images in Figures 3-5 are too small to tell anything about. Use fewer images per block so they can be larger.
- Can the authors comment at all on why each dataset seems to have a different optimal number of times to be augmented?
- It seems to me that the parallel GAN setup will have memory and compute requirements of approximately 8x (if the 8 GANs are being run in parallel). This is pretty substantial overhead. Are there any details I'm missing that would mitigate that?

---

### Decision · Program_Chairs · 2019-04-08
**Acceptance Decision**

Reject